# Mesenchymal Stem/Stromal Cells in Three-Dimensional Cell Culture: Ion Homeostasis and Ouabain-Induced Apoptosis

**DOI:** 10.3390/biomedicines11020301

**Published:** 2023-01-21

**Authors:** Alla N. Shatrova, Alisa P. Domnina, Natalia A. Pugovkina, Larisa L. Alekseenko, Irina I. Marakhova

**Affiliations:** 1Laboratory of Intracellular Membranes Dynamic, Institute of Cytology, Russian Academy of Sciences, Tikoretzky Ave, 4, 194064 Saint-Petersburg, Russia; 2Laboratory of Intracellular Signaling, Institute of Cytology, Russian Academy of Sciences, Tikoretzky Ave, 4, 194064 Saint-Petersburg, Russia

**Keywords:** Na/K pump, potassium, mitochondrial membrane potential, ouabain, apoptosis, three-dimensional spheroids, human endometrial mesenchymal stem/stromal cells

## Abstract

This study describes the changes in ion homeostasis of human endometrial mesenchymal stem/stromal cells (eMSCs) during the formation of three-dimensional (3D) cell structures (spheroids) and investigates the conditions for apoptosis induction in 3D eMSCs. Detached from the monolayer culture, (2D) eMSCs accumulate Na^+^ and have dissipated transmembrane ion gradients, while in compact spheroids, eMSCs restore the lower Na^+^ content and the high K/Na ratio characteristic of functionally active cells. Organized as spheroids, eMSCs are non-proliferating cells with an active Na/K pump and a lower K^+^ content per g cell protein, which is typical for quiescent cells and a mean lower water content (lower hydration) in 3D eMSCs. Further, eMSCs in spheroids were used to evaluate the role of K^+^ depletion and cellular signaling context in the induction of apoptosis. In both 2D and 3D eMSCs, treatment with ouabain (1 µM) results in inhibition of pump-mediated K^+^ uptake and severe K^+^ depletion as well as disruption of the mitochondrial membrane potential. In 3D eMSCs (but not in 2D eMSCs), ouabain initiates apoptosis via the mitochondrial pathway. It is concluded that, when blocking the Na/K pump, cardiac glycosides prime mitochondria to apoptosis, and whether a cell enters the apoptotic pathway depends on the cell-specific signaling context, which includes the type of apoptotic protein expressed.

## 1. Introduction

Mesenchymal stem/stromal cells (MSCs), multipotent clonogenic cells of mesodermal origin, can be isolated from numerous adult tissues, including the endometrium (eMSCs) [1,2,3,4,5,6,7]. MSCs’ differentiation potential, ease of harvest, low immunogenicity, and capacity to promote cell migration make them an attractive therapeutic remedy [8,9]. It is considered that they promote the survival and proliferation of host cells through paracrine activity as well as reducing inflammation [10,11]. The majority of clinical trials use the cellular suspension of MSCs as the therapeutic agent. However, the latest research suggests using the MSCs incorporated into three-dimensional (3D) structures [12,13,14]. Different 3D models of cell growth, such as scaffolds based on different synthetic or natural materials and seeded with cells, as well as scaffold-free models and cell spheroids are under development [15]. Spheroids, originally emerged as 3D aggregates of tumor cells, have long been used as a model for studying the tumors and their microenvironment, as well as for testing antitumor drugs [16]. Now this model of cell growth is used for the cultivation of MSCs isolated from different tissues [17,18,19,20,21].

How cells survive in spheroids is widely discussed. Single suspended cells are not capable to survive, however, in compact spheroids with developed cell–cell contacts, cells remain viable. Culturing of MSCs in 3D configuration leads to their phenotype shifts and acquirement of the features unusual for their two-dimensional (2D) cultures [15,22]. Some studies have shown that in spheroids, the cell–cell contacts make them responsive to the survival factor of cells [23]. Cultivation in spheroids augments the angiogenic potential of MSCs and enhances anti-inflammatory MSC properties [17,19,24,25]. As recently shown, eMSCs in spheroids are prone to apoptosis and in response to stress stimuli enter an apoptotic pathway [26]. It is hypothesized that basal down-regulation of anti-apoptotic and autophagy-related genes may provide the molecular basis for apoptosis-prone 3D eMSCs.

Changes in the molecular and functional properties of MSCs cultivated in spheroids open up new prospects for the clinical use of these cells. Currently, preclinical studies with the use of MSC spheres are conducted, aimed at the correction of various human diseases, such as skeletal system diseases, ischemic and cardiovascular disorders, and wound healing [27,28,29,30]. As shown recently, transplantation of eMSCs in spheroids can be used in the treatment of infertility in a rat model of Asherman’s syndrome and enhance skin wound healing in rats [21,31].

In our study, we paid attention to the high tendency of spheroids to the activation of the apoptosis program under stressors. In particular, cells in spheroids respond to damage induced by various stress insults by activating the apoptosis program, in contrast to monolayer cells in which a program of premature senescence is induced [26]. Starting from these observations, we studied the ion homeostasis as an indicator of cell viability during the transfer of eMSCs from a 2D monolayer culture to scaffold-free spheroids and tried to find out whether switching off the Na/K pump with ouabain can induce apoptosis in 3D eMSCs. The Na/K pump is the primary ion transport system responsible for the maintenance of the cell Na^+^ and K^+^ concentrations and the resting membrane potential. In cells, the Na/K pump is the most important homeostatic regulator. In recent years, cardiac glycosides have been suggested as drugs for eliminating cancer and senescent cells, however, the mechanisms of the selective killing activity of glycosides against dangerous cells remain poorly understood [32,33].

In the present study, we investigated the changes in ion homeostasis that accompany spheroid formation in the hanging drop technique from attached eMSCs in a monolayer and characterized the ion homeostasis of eMSCs cultivated in 3D configuration. To elucidate the mechanism of the apoptotic action of ouabain as a selective inhibitor of the ion pump, we studied the relationship between the intracellular content of K^+^ and the mitochondrial membrane potential in 2D and 3D eMSCs, and discussed the role of mitochondria priming and an enhanced apoptotic program in the induction of apoptosis by cardiac glycosides.

## 2. Materials and Methods

### 2.1. Cells and Experimental Design

All the experiments have been performed on human endometrial mesenchymal stem/stromal cells (eMSCs) established from a desquamated endometrium of menstrual blood from healthy donors [7]. Cells exhibited properties typical of mesenchymal stem/stromal cells: fibroblast-like morphology, multipotency, and expression of standard surface markers such as CD13, CD29, CD44, CD73, CD90, CD146, CD105 and are negative for the hematopoietic markers CD19, CD34, CD45, CD117, CD130. These cells are characterized by high rate of cell proliferation (doubling time 22–23 h). eMSCs were cultivated in DMEM/F12 medium (Gibco, Waltham, MA, USA) supplemented with 10% fetal bovine serum (FBS, HyClone, Logan, Utah, USA), 1% L-glutamine, and 1% penicillin-streptomycin (Gibco, Waltham, Massachusetts, USA). Cells were maintained at 37 °C, 5% CO_2_, and were subcultured twice a week at the split ratio of 1:3. For experiments, cells were harvested by trypsinization and plated at a density of 15 × 10^3^ cells per cm^2^ for experiments with monolayer cultures or were used to organize spheroids.

The procedures involved human cells were performed in accordance with the standards of the Declaration of Helsinki (1989) and approved by the Institute of Cytology Ethics Committee. All cell donors signed an informed consent for voluntary participation.

### 2.2. Spheroid Formation

Spheroids (three-dimensional cultures, 3D) were formed from monolayer eMSCs (2D cultures) using the hanging drop technique. First, 7.5–30 × 10^3^ cells in suspension per 30 µL of DMEM/F-12 medium containing 10% FBS were placed in drops on the cover of 10-cm Petri dishes (Corning, NY, USA) and inverted. During the next 24 h, in hanging drops, cells spontaneously aggregated and formed compact spheroids. For experiments, spheroids in hanging drops were directly treated with ouabain, or they were transferred to dishes coated with 2-hydroxyethyl methacrylate (HEMA, Sigma-Aldrich, St. Louis, MO, USA) and cultured in 2 mL of full growth medium with additions for the next 24 or 48 h. HEMA was used as a nonadhesive substrate in the preparation of cell spheroids in suspensions.

For flow cytometry assay, the single-cell suspension of spheroids was obtained. For this purpose, spheroids with culture medium were transferred to a 15 mL conical tube and centrifuged for 3 min at 1500 rpm. After centrifugation, the supernatant was removed; 10 mL of PBS solution was added, and the tube was centrifuged again. Then, the PBS solution was removed, and spheroids were incubated with 0.05% trypsin/EDTA solution for 5 min at +37 °C. After the incubation was completed, the cell suspension was gently pipetted 2–3 times to achieve spheroid dissociation. Finally, a three-fold volume of the culture media was added for trypsin inactivation.

### 2.3. Analysis of Cell K^+^ and Na^+^ Content and K^+^ Influx

Measurements of ions were performed essentially as described in detail previously [34,35]. To evaluate K^+^ influx, Rb^+^ was used as the physiological analog of K^+^. RbCl (final concentration 5 mM) was introduced into the culture medium for 20 min. To evaluate K^+^ influx via the Na^+^, K^+^-ATPase pump, prior to RbCl, 10^−4^ M ouabain (Sigma-Aldrich, St. Louis, MO, USA) was added to culture medium. Then, cells were rapidly washed 5 times with ice-cold isotonic MgCl_2,_ and cations were extracted with 1 mL of 1% trichloroacetic acid (TCA). TCA extracts were analyzed for Rb, K, and Na by emission flame photometry on a Perkin-Elmer AA 306 spectrophotometer with standard TCA solutions. TCA precipitates were dissolved in 0.1 N NaOH and analyzed for protein by the Lowry procedure, with serum bovine albumin as a standard. Ouabain-sensitive Rb^+^ uptake was calculated as the differences between the mean values measured in samples incubated with and without ouabain. The intracellular ion content was expressed as number of ions per amount of protein in each sample analyzed.

### 2.4. Flow Cytometry Assay of Apoptosis and Mitochondrial Membrane Potential

All samples were analyzed with a CytoFLEX flow cytometer (Beckman Coulter, Brea, CA, USA) or CytoFLEX S flow cytometer (Beckman Coulter, Brea, CA, USA), using CytExpert software (version 2.0, Brea, CA, USA). The data obtained are presented as diagrams, dot plots, and histograms.

To perform cell cycle analysis, 2D and 3D cells were harvested with 0.05% trypsin-EDTA solution, suspended in the fresh medium, permeabilized with 0.1% Triton X-100 (Sigma-Aldrich, St. Louis, MO, USA), and stained for 5 min with 2 µg/mL of 4’,6-diamidino-2-phenylindole (DAPI, Sigma-Aldrich, St. Louis, MO, USA). Cell cycle phase distribution was measured with CytoFLEX S flow cytometer and analyzed using CytExpert 2.0 software” (Beckman Coulter, Brea, CA, USA, 405 nm laser).

Apoptosis was assayed using Annexin V/Alexa Fluor™ 647 conjugate in accordance with the manufacturer’s instructions (Thermo Fisher Scientific, Waltham, MA, USA). Treated and untreated cells were harvested by trypsinization, washed with PBS, pelleted by centrifugation, and adjusted to a concentration 1 × 10^6^ cells/mL in 1× Annexin binding buffer (10 mM HEPES, 140 mM NaCl, 2.5 mM CaCl_2_, pH 7.4). Next, 1 × 10^5^ cells (100 μL of cell suspension) were stained with 5 μL of Annexin V, Alexa Fluor™ 647 conjugate (Thermo Fisher Scientific, Waltham, MA, USA), and 2 μL (final concentration 2 μg/mL) of DAPI for 15 min in the dark at room temperature. Then, 400 μL of 1× buffer was added to each sample, gently vortexed, and analyzed by flow cytometry as soon as possible.

For detection of the caspase activity, 6 or 24 h after the ouabain treatments, 3D cells were trypsinized, suspended in the growth medium, stained with Cell Event Caspase-3/7 Green Detection Reagent (Thermo Fisher Scientific, Waltham, MA, USA) for 20 min at 37 °C, and analyzed with a CytoFLEX flow cytometer (Beckman Coulter, 488 nm laser).

Tetramethylrhodamine (TMRM; Invitrogen, Carlsbad, CA, USA) dye was used as mitochondrial membrane potential (MMP) indicator. TMRM, as a cationic and lipophilic fluorescent dye, was used to detect changes in the membrane potential by tracking the redistribution of the dye. A healthy cell will display a robust fluorescent signal; however, the fluorescence diminishes when the mitochondrial membrane depolarizes due to changes such as induction of apoptosis. The protonophore CCCP (carbonyl cyanide m-chlorophenyl hydrazone), an inhibitor of mitochondrial oxidative phosphorylation, was used to dissipate the membrane potential and to define the baseline for the analysis of mitochondrial potential by TMREM fluorescence. Briefly, 50 mM CCCP (1000× solution) was added to cells at least 10 min prior to TMRM staining. Then, 100 µM TMRM (1000× solution) was added to all samples, incubated for 30 min at 37 °C, and analyzed by FACS. TMRM was excited with a 561 nm laser and analyzed in PE channel.

### 2.5. Quantitative RT-PCR

To analyze gene expression, total RNA was isolated with an Aurum™ Total RNA Mini Kit (BioRad, Hercules, CA, USA) according to the manufacturer’s instructions. RNA was quantified using NanoDrop ND-1000 Spectrophotometer (NanoDrop Technologies, Inc, Wilmington, DE, USA). cDNA was obtained via the reverse transcription of RNA using RevertAid H Minus First Strand cDNA Synthesis Kit (Thermo Fisher Scientific, Waltham, Massachusetts, USA) according to the manufacturer’s instructions. For qRT-PCR, cDNA was amplified with specific primers, using qPCRmix-HS SYBR (Evrogen, Moscow, Russia) in the BioRad CFX Opus-96 real-time system (BioRad, Hercules, CA, USA) according to the kit’s enclosed protocol. The volume of RT and PCR reactions was 20 μL. Expression of target genes was normalized to GAPDH or actin gene. Primers and reaction conditions are presented in Table 1.

### 2.6. Statistical Analysis

Results are presented as the mean with standard error of the mean from at least three independent experiments. Statistical significance was assessed using either Student’s t-test in case of pair comparisons or one-way ANOVA with post-hoc Tukey HSD test was used to determine the significance of differences among groups.

## 3. Results

### 3.1. Intracellular K^+^ and Na^+^ Content and Rb^+^ (K^+^) Influxes during eMSCs Transit from 2D Monolayer Culture to 3D Spheroids

Experiments were designed to obtain eMSCs as spheroids by the hanging drop method and to determine the intracellular content of K^+^ and Na^+^ and the short-term Rb^+^ uptake (as a measure of K^+^ influx) during spheroid formation and their survival, as well as after the dissociation of spheroids, followed by seeding into a monolayer (2D) culture. As seen in Figure 1A, when detached from the monolayer culture by tripsinization (0.05% trypsin, HyClone) and thereafter washed extensively with DMEM/F12 growth medium containing 10% fetal bovine serum for 2–3 min, eMSCs have high K^+^ content, which is within the range of cell K^+^ content in a pre-confluent 2D culture. However, in detached cells in suspension, the Na^+^ content was increased so much that it was comparable to the intracellular K^+^ content and the cell K/Na ratio approached 1 (Figure 1A).

We asked whether the elevated Na^+^ content in detached eMSCs was a consequence of cell treatment with trypsin solution which is used to remove cells from adhesive plates and which is able to increase the permeability of cell membrane to Na^+^ [36]. To answer this question, we maintained the detached eMSCs in suspension in growth medium by soft rocking up to 2 h in a humidified chamber with 5% CO_2_ at 37 °C. It turned out that under these conditions, an increased content of Na^+^ (and high K^+^ content) in eMSCs persisted as long as the cells were in suspension (Figure 1A). When the suspended eMSCs were plated into 2D culture, intracellular Na^+^ gradually decreased, approaching a lower, normal value the next day after plating (Figure 1A). These data suggest that in detached eMSCs in suspension, the elevated Na^+^ content and low K/Na ratio is not due to trypsin treatment, but rather due to cell detachment and impaired cell-extracellular matrix and cell–cell contacts.

The normal ion homeostasis with high K/Na ratio is also restored when dissociated eMSCs in suspension are organized into spheroids in hanging drops (Figure 1B). One day after the eMSCs suspension was placed into hanging drops, cell Na^+^ content decreased to 100–200 µmol/g, which is typical for growing cell cultures in 2D conditions. At the same time, the content of K^+^ in cells does not change significantly and amounts to 500–600 µmol/g. According to our previous data, such lower values of cell K^+^ content per 1 g of cell protein (K^+^/protein ratio in a cell) are characteristic of non-cycling and quiescent cells [37]. In this study, eMSCs spheroids contain 94 ± 2% G0/G1 cells, 1 ± 0.2% S cells, 3 ± 0.52% G2/M cells (instead of 81 ± 0.15% G0/G1 cells, 7 ± 0.3% S cells, 13 ± 0.16% G2/M cells in 2D culture) so eMSCs in 3D culture represent arrested cells (Figure 1D,E). It is interesting that the dissociation of eMSCs spheroids with a trypsin-containing solution, followed by short-term (three times) washing with a culture medium that does not contain trypsin, is accompanied by a sharp drop in the K/Na ratio in the cells to 0.5 as a result of the loss of K^+^ and the accumulation of Na^+^ (Figure 1B). When thus dissociated from spheroids, eMSCs are placed on an adhesive surface, where they attach and begin to proliferate. Under these conditions, the transition from dissociated spheroids to 2D cell culture is associated with an increase in the content of K^+^ in cells and the beginning of cell culture growth (Figure 1B).

To evaluate Na/K pumping activity in 3D cell culture, short-term, ouabain-inhibitable Rb^+^ uptake was determined as a measure of pump-mediated K^+^ influx. In 3D eMSCs, ouabain-inhibited Rb^+^ influx accounts for a significant part of the total Rb^+^ uptake and somewhat higher than that in 2D eMSCs (Figure 1C). To determine whether the increased ouabain-inhibitable Rb influx in 3D eMSCs is proportional to changes in intracellular Na^+^ content, we compared pump rate coefficients calculated as the ratio of ouabain-inhibitable Rb^+^ uptake to intracellular Na^+^ content in 2D and 3D cultures [38,39]. It turned out that the pump rate coefficients are close in value both in eMSCs spheroids (0.021 ± 0.002 min^−1^) and in preconfluent eMSCs monolayer culture (0.025 ± 0.002 min^−1^), which indicates that the increased ouabain-inhibitable Rb^+^ transport in spheroids is not associated with a change in the intrinsic properties of the Na/K pump, but is a consequence of flux concentration relations in existing ion pumps. We observed that pump-mediated Rb^+^ influx remained high and stable for 2 days of cell survival in spheroids.

In 3D eMSCs cultivated under hanging drop conditions, a high K/Na ratio and normal contents of K^+^ and Na^+^ persist for at least 2 days at a cell concentration of (7.5–30) × 10^3^ cells per 30 µL of culture medium. High ion gradients are also maintained if 2-day-old spheroids were placed on non-adherent HEMA-coated dishes and further cultured in a humidified chamber with 5% CO_2_ at 37 °C for the next 2 days.

Taken together, our findings indicate that eMSCs, organized as compact spheroids in hanging drops, represent non-cycling but functionally viable cells that are characterized by an active Na/K pump and high intracellular K/Na ratio. A peculiar feature of eMSCs spheroids is a lower K^+^ content per g cell protein as compared to growing monolayer cultures of eMSCs. As to detached from monolayer and suspended cells as well as dissociated spheroids, it turned out that the loss of cell–cell and cell-extracellular matrix contacts leads to a reversible dissipation of ion gradients between the cell and the extracellular medium.

### 3.2. Ouabain Induced Apoptosis in eMSCs Spheroids

As recently shown, eMSCs spheroids activate the apoptotic program in response to various stressors, thus being committed to programmed cell death [26]. We addressed this property of eMSCs in a 3D configuration to elucidate the ability of cardiac glycosides to induce apoptosis. In our recent study, we tested the ability of ouabain to inhibit the ion pump in proliferating eMSCs and found that at a concentration of 10^−7^ M, more ouabain stops ion pumping, thus leading to the disruption of cell K/Na gradients [40]. Therefore, in this study, we applied ouabain at a concentration of 10^−6^ M to discover the role of K^+^ depletion in apoptosis induction by cardiac glycosides. As detected by flow cytometry assay, in the 3D eMSCs population, the proportion of AnnV+/DAPI− cells (defined as early apoptotic cells) increased after treatment with 1 µM ouabain (Figure 2A and Figure 3Aa). Apoptosis induction was also seen in 3D MSCs, organized as spheroids, and then cultured on HEMA-coated dishes in complete growth medium (2 mL) with 1 µM ouabain for the next 24 or 48 h (Appendix A). In contrast to spheroids, apoptosis tests performed in ouabain-treated 2D eMSCs did not reveal signs of activation of the apoptotic program according to the AnnV+ test, although the number of DAPI+ (“dead”) cells in ouabain-treated cells in monolayer culture increased (Figure 2B and Figure 3Ba,b).

We analyzed how cell sizes change in 2D monolayers and 3D spheroids after treatment with ouabain. A decrease in flow cytometric forward light scatter (FSC) is commonly interpreted as a sign of apoptotic cell volume decrease: indeed, a delayed decrease in FSC occurs along with the accumulation of annexin-positive cells [41,42]. Flow cytometric analysis of eMSCs in spheroids showed that ouabain treatment decreased FSC, indicating a decrease in cell size and confirming ouabain-induced apoptosis in eMSCs organized as spheroids (Figure 3Ac,C). On the contrary, in the monolayer, the size of ouabain-treated eMSCs (as judged by FSC increase) gradually increased, which was in favor of the necrotic (or mixed) type of death of these cells (Figure 3Bc).

Ouabain-induced apoptosis in 3D eMSCs is associated with an increased caspase 3/7 activity (Figure 3D). It should be mentioned that untreated eMSCs spheroids—the control—showed the presence of caspase 3/7 activity, which was mainly associated with dead cells: 3D eMSCs in hanging drops usually contain 4–18% DAPI+ (“dead”) cells. The activation of caspase 3/7 in ouabain-treated 3D eMSCs indicates the transit to the executive stage of apoptosis and irreversibility of induced cell death. Overall, flow cytometry data on annexinV induction and caspase 3/7 activation indicate that ouabain initiates apoptosis in eMSCs cultured in spheroids but not in a monolayer.

To confirm the possibility of switching on the apoptotic program in 2D and 3D eMSCs after exposure to ouabain, we analyzed changes in the expression of the proapoptotic *BAX* gene, which promotes the release of cytochrome C, leading to the activation of caspases, and the antiapoptotic gene *BCLXL*, which prevents the release of cytochrome C from mitochondria. The results of our studies showed that the baseline level of *BAX* in 3D eMSCs is higher than in 2D cell culture (Figure 4). Twenty-four h after exposure to ouabain, there is no change in the expression of the pro-apoptotic *BAX* gene in either 2D or 3D eMSCs. In contrast to 2D eMSCs, where the expression of the anti-apoptotic *BCLXL* gene remains unchanged, it significantly decreases in 3D eMSCs (Figure 4). It can lead to alterations in the balance between pro- and antiapoptotic genes and the activation of the apoptosis program.

### 3.3. Effect of Ouabain on Cellular K^+^ Content and Mitochondrial Membrane Potential in 2D and 3D eMSCs

It is suggested that ouabain can initiate apoptosis via a mitochondria-dependent pathway [43]. In the following set of experiments, we tested how a decrease in cytoplasmic K^+^ in ouabain-treated cells affects the mitochondrial membrane potential (MMP), which is critical for mitochondria to produce ATP and normal mitochondria functioning. In the same batches of cells with ouabain-suppressed ion pumps, we measured the internal K^+^ and Na^+^ content, pump-mediated Rb^+^ influx (as a measure of Na/K pump activity), and MMP. The results showed that in both 2D and 3D, eMSCs 1 µM ouabain rapidly inhibited pump-mediated K^+^ influx as evidenced by the drop in ouabain-sensitive Rb^+^ uptake in ouabain-treated cells (Figure 5B,D).

Within 1 h after the addition of ouabain, the cells lose most of their internal K^+^, and after 24 h, the K^+^ content in the cells is only 100–200 µmol/g (Figure 5A,C). The study of the relationship between the dose of ouabain and the content of K^+^ in cells, carried out on 2D eMSCs, showed that, although the process of depletion of intracellular K^+^ depends on the concentration of ouabain and the time of its action, in cells with the ion pump turned off, by 24 h, the same low content of K^+^ is established in both 2D and 3D eMSCs (Figure 5A,C).

To investigate the effect of intracellular K^+^ depletion on mitochondrial function, we examined the changes in mitochondria membrane potential (MMP) in ouabain-treated eMSC spheroids. MMP is an indicator of mitochondrial activity because it reflects the process of electron transport and oxidative phosphorylation. Oubain (1 μM) was added directly to the hanging drops with organized spheroids, after which MMP was assayed by flow cytometry measurements of TMRM fluorescence after 1, 6, or 24 h of ouabain addition. Figure 6A shows that ouabain treatment leads to time-dependent changes in TMRM fluorescence. A drop in fluorescence was observed 1 h after the addition of ouabain to hanging drops with spheroids; thereafter, fluorescence remained low for 24 h (Figure 6B). The data in Figure 6A,B revealed similar changes in MMP in ouadain-treated eMSCs cultivated in both 3D and 2D configurations. These results demonstrate a strong and rapid depolarization of mitochondria in K^+^-depleted cells, thus indicating a loss of mitochondria function.

## 4. Discussion

The human eMSCs incorporated into various 3D structures are more appropriate in clinical trials, however, the cell phenotype and cell behavior are changed as the culture environment changes from 2D to 3D configuration [16,22,23,24,25,26]. In the present study, we revealed peculiar changes in ion homeostasis during the transition from 2D to 3D cell culture and the formation of eMSCs spheroids.

Detached eMSCs accumulate Na^+^ and have dissipated transmembrane ion gradients, while in spheroids, eMSCs restore the low content of Na^+^ and the high K/Na ratio characteristic of functionally active cells. MSCs organized as spheroids are non-proliferating cells with a lower K^+^ content per g of cell protein, which is typical for quiescent cells and may mean lower water content in cells cultured in 3D scaffold-free conditions. Non-cycling eMSCs in spheroids are viable cells that begin cycling when seeded into a monolayer culture. It is noteworthy that cell adhesion in 2D culture and the transition to proliferation are associated with an increase in the K^+^ content per g of protein.

Changes in the ion homeostasis of eMSCs during 2D–3D transit are likely the result of changes in cell-extracellular matrix (ECM) and cell–cell contacts. The spheroids in hanging drops represent a scaffold-free culture system in which cell–ECM contacts are destroyed. Several studies have reported that spheroids represent spatial organization of cells with enhanced cell–cell interactions, and E-cadherin is a key molecule in the formation of cell–cell contacts in the three-dimensional organization of cells [44,45,46,47,48]. It is also established that the Na^+^,K^+^-ATPase promotes adhesion and the formation of tight junctions due to the self-adhesive properties of the β-subunit, and E-cadherin binds to the β-subunit, forming a complex for adhesion [49,50,51]. Recent studies also suggest that the different isoforms of the β-subunit of Na,K-ATPase are implicated in regulating cell motility during cancer progression [52,53]. We propose that cell–cell interactions during spheroid formation can contribute to the optimization of ion pump activity and the maintenance of transmembrane gradients of Na^+^ and K^+^, which are necessary for the normal functioning of eMSCs—in particular, to enhance the paracrine activity of cells.

A peculiar feature of eMSCs in 3D spheroids is a lower K^+^ content per g cell protein (500–600 µmol/g) as compared to eMSCs in pre-confluent monolayer culture with high rate of proliferation (700–900 µmol/g). Recently, we found a decreased cell K^+^/protein ratio during the development of senescence in stress-induced eMSCs [40]. Human eMSCs in compact multicellular spheroids as well as senescent eMSCs in monolayer culture are arrested, non-proliferating cells, and a lower cell K^+^/protein ratio is consistent with our previous finding that a decrease in this index indicates a decrease in cell proliferation [35,54]. In a recent study, we showed that stimulation of quiescent human lymphocytes with mitogens is accompanied by an increase in both cell K^+^ content per gram of cell protein and cell water content per gram of cell protein, so that as activated human lymphocytes progress from quiescence towards proliferation, their K^+^ concentration remains constant [37]. Together, these data shed light on the role of K^+^ in the regulation of cell proliferation. Based on our experimental studies and previous theoretical considerations about the ion–water balance in animal cells [55,56], we conclude that K^+^ is involved in the regulation of cell proliferation as the dominant intracellular osmolyte that controls changes in cellular volume and water content in cells. It is important to emphasize here that, since a high ratio of K^+^ content to protein mass in the cell indicates an increased content of water in the cell, it can be assumed that the cycling cell is more hydrated than the resting one.

In light of these findings, the lower K^+^/protein ratio in 3D eMSCs is consistent with their loss of proliferative activity. Indeed, when eMSCs are seeded from spheroids into a 2D culture, they begin to cycle and the K^+^/protein ratio increases. By analogy with the cases when measurements of the ion content were carried out with a parallel measurement of the water content [37,57], we assume that a lower K^+^/protein ratio in 3D eMSCs may indicate a lower water content in these cells, that is, lower hydration of non-cycling cells in 3D spheroids compared to proliferating cells in 2D monolayer culture. This assumption is consistent with a recent direct measurement of intracellular dry mass density: using ratiometric stimulated Raman scattering microscopy to measure the intracellular mass density of single cells, it was found for the first time that 3D spheroids of HEK293 cells have a higher density than cells in 2D monolayer culture [58].

The functional significance of changes in cell hydration has not been sufficiently explored. There is evidence that reduced cellular hydration can directly affect macromolecule concentration and macromolecular crowding and impact cellular metabolism, signaling, and gene activity [59,60,61,62,63].

Adhesion receptors not only support a physical attachment between ECM and the cytoskeleton, but they also generate an adhesion-dependent signaling via a number of adaptor proteins and kinases, and the disruption of connection with the extracellular matrix leads to a specific type of apoptosis known as anoikis [64,65,66]. That is why recent data that eMSCs in spheroids become commitment for apoptosis are not surprising [26]. In the present study, we paid attention to this peculiarity of eMSCs in 3D configuration to elucidate the mechanism by which ouabain, a selective inhibitor of the Na/K pump, could initiate apoptosis: at present, cardiac glycosides are suggested as drugs for targeting the cancer or aging cells [32,33].

In animal cells, the Na/K pump is the primary ion transport system responsible for the maintenance of the low Na^+^ and high K^+^ concentrations and the resting membrane potential. Disruption of transmembrane ion gradients causes multiple effects on the cell. Steep Na^+^ and K^+^ gradients are used to facilitate secondary transport of molecules (sugars, aminoacids, metabolites) and other ions (H^+^, Ca^2+^, Cl^−^). Selective pump inhibition leads to the cytoplasm acidification and the increases in intracellular Ca^2+^ concentration. Further, glycosides, when bound to Na^+^, K^+^-ATPase, trigger several signaling pathways, including the Src-EGFR-Ras-Raf-MAPK signaling [67,68,69,70,71]. An analysis of transcriptomic changes in ouabain-treated cells permits to reveal a set of upstream Na^+^/K^+^-sensitive genes, among which were the early immediate genes [72]. Recent studies have also identified intracellular targets for cardiac glycosides, the modulation of which may not be related to the inhibition of membrane Na^+^,K^+^-ATPase. These include endosomal trafficking of Na^+^,K^+^-ATPase and initiating the signaling pathways that are involved in Src-mediated cell death pathways [73,74]. In view of the multiplicity of intracellular pathways for cardiac glycosides, it is not surprising that the type of induced death is dependent on the type of cell. Numerous studies demonstrated tissue- and species-dependent actions of cardiac glycosides on cell survival: in addition to apoptosis, these drugs cause necrosis, pyroptosis, autophagic, or hybrid cell death.

Human eMSCs are highly resistant to apoptosis [75,76,77]. In the present study, we found that when organized into spheroids, eMSCs—in response to ouabain—initiate apoptosis via a mitochondrial- and caspase-dependent pathway, while eMSCs cultured as monolayers do not. As recently shown, eMSCs in spheroids are prone to apoptosis and, in response to stress stimuli, enter an apoptotic pathway due to the down-regulation of anti-apoptotic Bcl-xL gene and up-regulation of pro-apoptotic Bax and PUMA genes [26]. Within the present study, we revealed that in contrast to 2D eMSCs, in ouabain-treated 3D eMSCs, the expression of the anti-apoptotic *BCLXL* gene significantly decreased. It can lead to alteration in the balance between pro- and anti-apoptotic genes and the activation of the apoptosis program. Studies with some cancer cells also demonstrate that the ability of ouabain to initiate apoptosis correlates with reduced expression of the anti-apoptotic proteins Mcl-1, Bcl-XL, and Bcl-2, or increased expression of the pro-apoptotic proteins Bid and Bax [78,79,80]. The present study provides insight into how alterations in cell signaling context may influence the ability of ouabain to induce apoptosis: in cells with an ouabain-inhibited Na/K pump, the necessary intracellular context to induce apoptosis would include the enhanced pro-apoptotic program.

Another aspect of ouabain-induced apoptosis in eMSCs spheroids concerns the role of intracellular K^+^ depletion. In our experiments, blockade of the Na/K pump with ouabain results in a rapid (within the first hour) decrease in K^+^ content in human eMSCs, grown in both 2D and 3D configurations. Excessive K^+^ efflux and intracellular K^+^ loss are key early events in receptor-induced, extrinsic apoptosis [81,82,83]. Here, we focused on long-term depletion of intracellular K^+^ caused by ion pump inhibition and hypothesized that by reducing cellular K^+^, ouabain treatment could prime mitochondria for intrinsic apoptosis that is triggered by stressors.

Mitochondrial function and architecture are fundamentally dependent on the concentrations of key cellular ions, K^+^, H^+^, Ca^2+^ [84]. Under normal physiological conditions, the high electric potential difference generated by the proton pump across the inner mitochondrial membrane, determines the K^+^ influx into matrix. Changes in the matrix volume due to this K^+^ influx and concomitant water entry are compensated by the activity of the electroneutral K^+^/H^+^ antiporter [85].

In this study, for the first time, we carried out parallel direct measurements of K^+^ content and mitochondrial membrane potential in eMSCs treated with ouabain at concentrations that inhibit the Na/K pump and initiate apoptosis, and showed that K^+^ depletion leads to the breakdown of mitochondrial potential, thus priming the mitochondria to apoptosis. When mitochondrial potential drops, changes in the mitochondria volume and architecture facilitate cytochrome *c* release into intra-mitochondrial space [86,87,88]. Further, to release the cytochrome *c* into the cytoplasm, a voltage-gated anion channel (VDAC), the function of which is directly regulated by Bcl-2 proteins, must be opened: it is proposed that pro-apoptotic BAX interacts with VDAC, thus forming a channel through which cytochrome *c* can pass into the cytoplasm and participate in the formation of apoptosomes [89,90,91,92,93]. Thus, in ouabain-treated, mitochondria-primed cells, the active pro-apoptotic proteins are required to induce irreversible apoptotic events.

## 5. Conclusions

Organized as free spheroids, eMSCs are viable but non-proliferating cells with an active Na/K pump and a lower K^+^ content per g cell protein, which is typical for quiescent cells, and which means lower hydration of eMSCs in 3D configuration. In contrast to monolayer culture, eMSCs in spheroids are committed to apoptosis, and in response to ouabain at concentrations which blocked the ion pump, they initiate the apoptotic program via mitochondrial pathways. In cells with the Na/K pump turned off, a sharp decrease in cytoplasmic K^+^ content leads to disruption of the mitochondrial membrane potential and mitochondria priming to apoptosis; the entry of such a cell into the apoptotic pathway depends on the cell-specific context, which includes the type of expressed Bcl-2 protein.

Taken together, the available data shed light on the possibility of applying the cardiac glycosides as drugs for targeting dangerous cells. Originally described to treat—in small, non-toxic doses—cardiovascular diseases, at present, cardiac glycosides in higher concentrations are suggested as anti-cancer and anti-aging drugs. As shown in present study, at doses that inhibit the Na/K pump and disrupt the high K/Na ratio, ouabain primes the cell for death and, depending on the specific signaling context, may promote further entry into the apoptotic pathway. However, the Na/K pump is the main homeostatic cellular regulator that ensures the stability and survival of animal cells, and cardiac glycosides, turning off the pump, destroy not only highly proliferating tumor or arrested senescent cells, but also normal cells. Thus, blockade of the ion pump by cardiac glycosides is unlikely be a reliable way to selectively kill dangerous cancer or senescent cells.

## Figures and Tables

**Figure 1 biomedicines-11-00301-f001:**
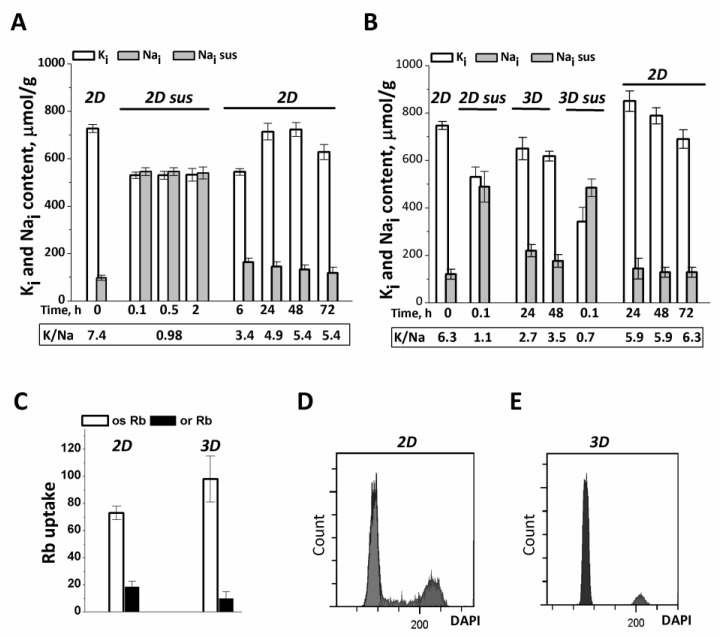
Ion homeostasis of eMSCs during the transition from 2D to 3D culture. (**A**) Na^+^ content increases in eMSCs detached from the monolayer culture (*2D*) and then maintained in suspension (*2D sus*). Two-dimensional eMSCs (passage 11) were taken for experiments as a pre-confluent monolayer was achieved; *2D sus*—cells from monolayer were harvested by trypsinization, cultivated as suspension in DMEM/F12 medium supplemented with 10% fetal calf serum for 2 h, and then grown as 2D culture (*2D).* (**B**) In spheroids (*3D*), eMSCs have a low Na^+^ content as compared to cells in suspension (*2D sus*). Three-dimensional eMSCs (passage 11) were grown in hanging drops at a cell density of 15 × 10^3^ cells/30 µL of growth culture medium and then seeded as a monolayer (*2D).* Ion content is presented as µmol/g protein. Data in (**A**,**B**) were obtained in two separate set of experiments. (**C**) Rb influxes, ouabain-inhibitable and ouaban-resistant, in 2D and 3D eMSCs. Rb^+^ influx is presented as µmol/g, 30 min. Data are shown as the mean ± SD, n = 3–5 from one experiment out of three conducted with the same protocol. (**D**,**E**) Flow cytometric analysis of the distribution of cell cycle phases in 2D (**D**) and 3D € eMSCs.

**Figure 2 biomedicines-11-00301-f002:**
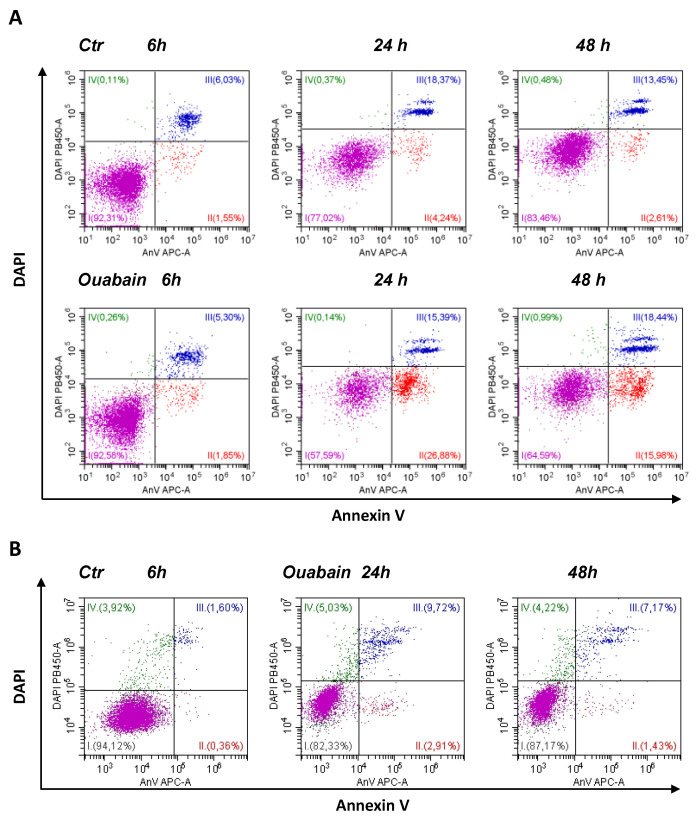
Ouabain induces apoptosis in eMSCs cultivated in three-dimensional spheroids rather than in a monolayer. (**A**) Apoptosis assay of eMSCs in spheroids obtained by the hanging drop method and exposed to ouabain. AnnexinV/DAPI flow cytometry cytograms (dot plots) measured in 3D eMSCs, control (Ctr) and treated with 1 µM ouabain for 6, 24, and 48 h. Data from one experiment out of four performed on the same protocol. (**B**) Annexin V/DAPI double staining with monolayer (2D) eMSCs, control and treatment with 1 µM ouabain for 24 and 48 h. AnV-/DAPI-live cells (I); early apoptotic AnV+/DAPI- cells (II); late apoptotic cells AnV+/DAPI+ (III). DAPI+AnV- necrotic cells (IV).

**Figure 3 biomedicines-11-00301-f003:**
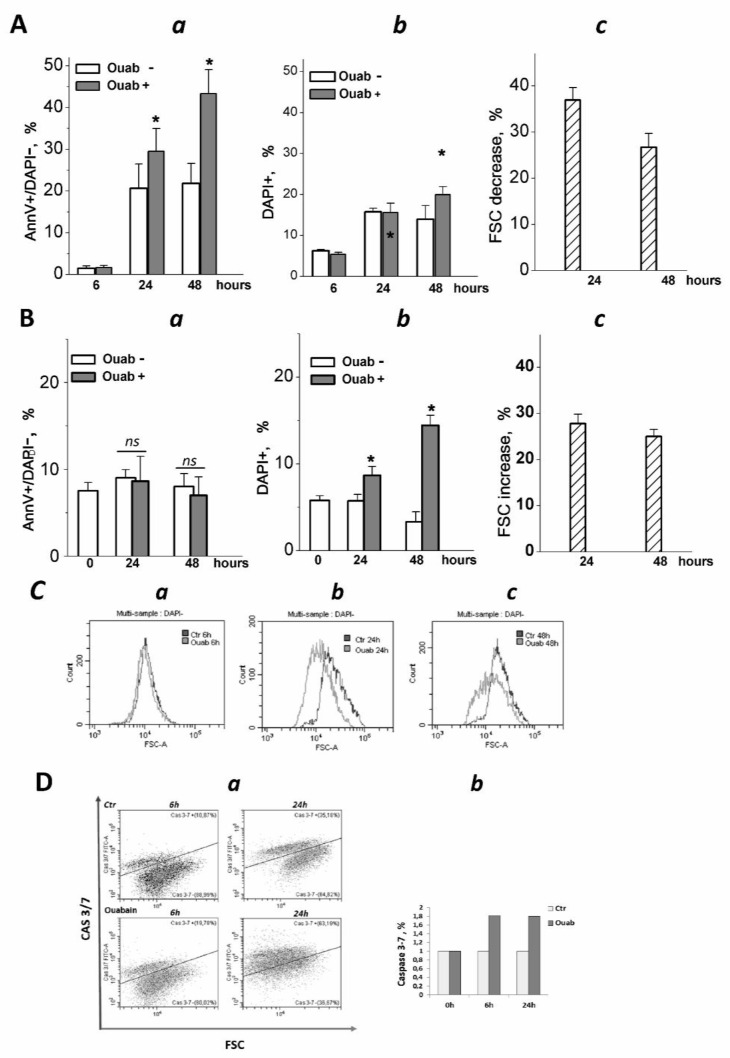
Three-dimensional eMSCs activate the apoptotic program in response to ouabain. (**A**,**B**) Quantification of apoptotic cell fractions in 2D and 3D eMSCs exposed to 1 µM ouabain; **a**—early apoptotic An+/DAPI- cell fraction; **b**—dead DAPI+ cell fraction; **c**—FSC decrease in ouabain-treated 3D eMSCs (**A**) and FSC increase in ouabain-treated 2D eMSCs (**B**), % to FSC of ouabain-untreated 3D (**Ac**) and 2D (**Bc**) cells. Data are presented as mean ± SD of four independent experiments performed in triplicate; significant difference between ouabain-treated and untreated cells at the same time of experiment was calculated using one-way ANOVA with Tukey’s post hoc tests, * *p* < 0.05; ns not significant. (**C**) Representative histograms showing FSC reduction in eMSCs spheroids exposed to 1 µM ouabain for 6 (**a**), 24 (**b**), and 48 (**c**) h. (**D**) Flow cytometry assay for 3/7 caspase activation in 3D eMSCs exposed to 1 µM ouabain reveals a cell fraction with increased caspase (**a**). (**b**) Quantification of the cell fraction with active caspase in 3D ouabain-treated eMSCs. Data from one experiment from two independent experiments on eMSCs in spheroids. Three-dimensional eMSCs (passage 10) were grown in hanging drops at a cell density of 7.5 × 10^3^ cells/30 µL of growth culture medium. Ouab—ouabain.

**Figure 4 biomedicines-11-00301-f004:**
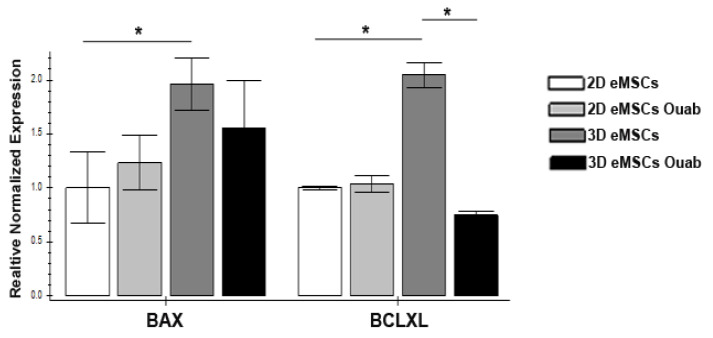
Ouabain-induced gene expression profiles in 2D and 3D eMSCs. Expression. Pro-apoptotic (*BAX*) and anti-apoptotic (*BCLXL*) genes in 2D and 3D eMSCs, controled and exposed to 1 µM ouabain (24 h after the ouabain treatment). Data are shown as mean ± SD (n = 3). * *p* < 0.05 the control 3D vs. control 2D eMSCs or the control 3D vs. the ouabain-treated 3D eMSCs.

**Figure 5 biomedicines-11-00301-f005:**
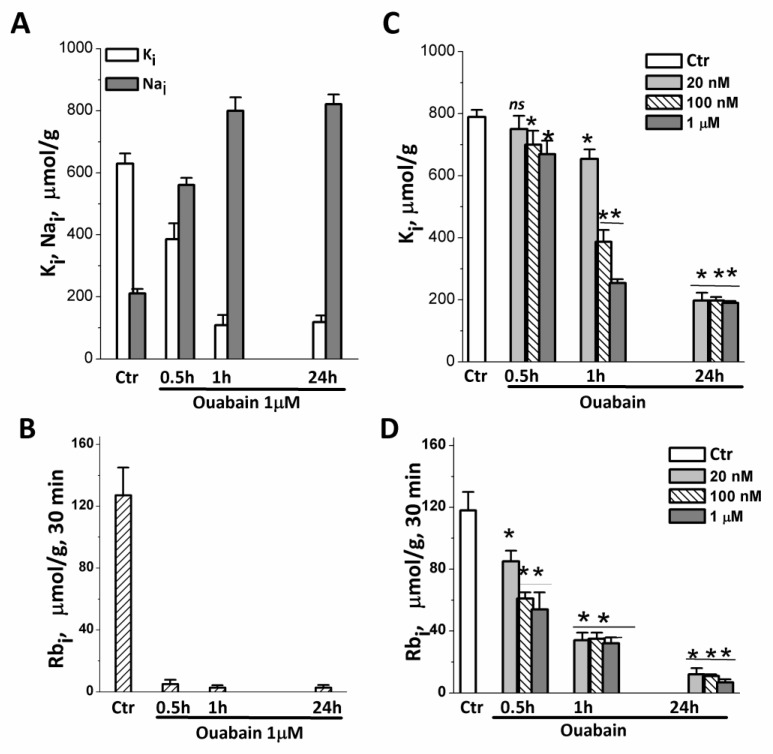
Changes in intracellular K^+^ and Na^+^ content and pump-mediated Rb^+^ uptake in eMSCs growing in spheroids and a monolayer after treatment with ouabain. (**A**,**B**) Cells in spheroids were treated with 1 μM ouabain for 0.5, 1 and 24 h and analyzed for K^+^ (black columbs), Na^+^ (light columns) content (**A**), and Rb^+^ uptake (**B**) by flame emission photometry. (**C**,**D**) Cells in monolayer were treated with 20, 100 nM, or 1 μM ouabain for 0.5, 1, and 24 h and analyzed for K^+^ content (**C**) and Rb uptake (**D**). Data are shown as mean ± SD (n = 3) of one representative experiment from two performed on the same scheme. * *p* < 0.5, ** *p* < 0.05, *** *p* < 0.005 vs. the control cell, ns—insignificant. Ctr—control cells.

**Figure 6 biomedicines-11-00301-f006:**
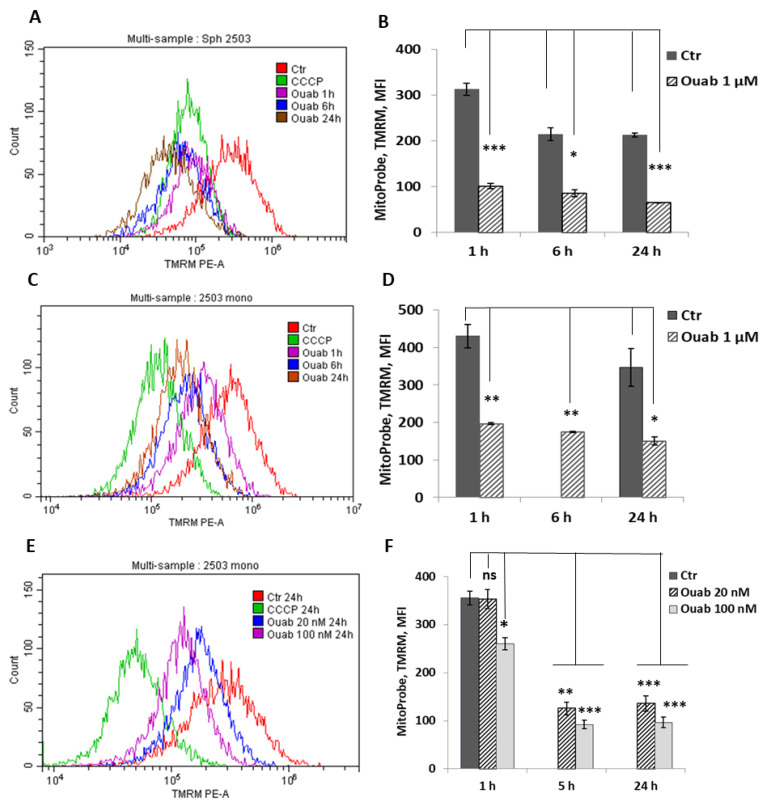
Ouabain-induced changes in the membrane potential of mitochondria in eMSCs growing in spheroids and in a monolayer. Cells in spheroids (**A**,**B**) and monolayer were treated with 1 μM ouabain (**C**,**D**) or 20 nM and 100 nM ouabain (**E**,**F**) and analyzed at the indicated time points using FACS. Changes in mitochondrial membrane potential (MMP) were assessed by TMRM (tetramethylrhodamine, methyl ester) fluorescence. The protonophore CCCP (carbonyl cyanide m-chlorophenyl hydrazone), an inhibitor of mitochondrial oxidative phosphorylation, was used to define the baseline for the analysis of MMP; 50 mM CCCP was added to probes with cells before FACS analysis. Data are shown as mean + SD (n = 3) from one experiment from two performed on the same scheme. * *p* < 0.5, ** *p* < 0.05, *** *p* < 0.005 vs. the control cells. Ctr—control cells. Ouab—ouabain.

**Table 1 biomedicines-11-00301-t001:** The primers and conditions for qRT-PCR.

Symbol	Primer Sequence	Amplification Conditions	PCR Product Size (bp)	Accession Number
BAX	F: 5′-GGGTTGTCGCCCTTTTCT-3′ R: 5′-CAGCCCATGATGGTTCTGATCAG-3′	93 °C, 20 s, 59–62 °C, 30 s 72 °C, 30 s	91	XM_047439168.1
Bcl-Xl	F: 5′-GCTTGGATGGCCACTTACCT-3′ R: 5′-GGGAGGGTAGAGTGGATGGT-3′	93 °C, 20 s, 59–62 °C, 30 s 72 °C, 30 s	231	NM_001317919.2
GAPDH	F: 5’-GAGGTCAATGAAGGGGTCAT-3′ R: 5’-AGTCAACGGATTTGGTCGTA-3′	93 °C, 20 s, 59–62 °C, 30 s 72 °C, 30 s	100	NM_001357943.2

## Data Availability

Data sharing not applicable.

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
