# Peer review of "Mesenchymal Stem/Stromal Cells in Three-Dimensional Cell Culture: Ion Homeostasis and Ouabain-Induced Apoptosis"

_biomedicines, 2023, doi:10.3390/biomedicines11020301_

Round 1
Reviewer 1 Report
The main result reported by Shatrova and coauthors is that eMSC cells organized into spheroids are more susceptible to ouabain-induced apoptosis than individual cells attached to a culture plate. This seems to be a useful result. The variety of methods employed adds value to the work. I do have a few qualms though, the main of which concerns the quality and organization of figures.
Fig. 1A. My guess is that this figure shows the Na and K concentrations first in attached cells, then after trypsinization, and then after reattachment. Do the black bars indicate sodium? If so, why show sodium bars in three different ways?
Fig. 2B. Part of this figure (2D cultures) seems to replicate Fig. 1A
Fig. 1C. What is os? If it is ouabain-inhibitable, then why os?
Fig. 1D. Were the cells fixed for DAPI staining?
Fig. 2. Each quadrant contains roman numerals followed by two more numbers in parentheses. What are they? The legend says (for example): “blue, late apoptotic AnV+/DAPI- dead cell fraction (II)” but that is not consistent across the graph, and the numbers in parentheses are nowhere explained.
Fig. 3. What is the difference between A and B? Why is one graph labeled “FSC increase” and the other “FSC decrease”?
Fig. 5. What is the difference with Fig. 1?
Fig. 6. What is MultiProbe? Does MFI stand for mean fluorescence channel? If A and B are for spheroids and C and D are for monolayer, what are E and F?
In some graphs, ouabain is abbreviated as ou, in others as ouab, and in yet others it is not abbreviated at all.
There are numerous distracting leftovers from the original files, such as small letters over the plots.
How were control (Ouab-) experiments conducted?
Other concerns and suggestions:
This is not the first study of apoptosis within spheroids (e.g., Korff, T., & Augustin, H. G. (1998). Integration of endothelial cells in multicellular spheroids prevents apoptosis and induces differentiation. The Journal of cell biology, 143(5), 1341-1352). The new results should be discussed in light of what is already known.
The authors seem to overstate their findings. As far as I can tell, they have only shown that ouabain induces apoptosis but not that it acts through potassium depletion. If they wish to prove a link between potassium and apoptosis, they could have complemented ouabain experiments with other ways to deplete intracellular potassium. Also, to maintain that apoptosis is caspase-dependent (line 518), caspase may have to be inhibited.
Presumably, the interior of spheroids has very little extracellular space. If the membrane of one cell faces the membrane of another with little intervening space, how does the membrane pump operate under such conditions? It would be interesting to have the authors’ thoughts on this question.
Line 302: “Ouabain treatment increased FSC, indicating a decrease in cell size”. This may be so, but such a controversial statement cannot be made in passing, without some discussion and references
Line 435. It takes some effort to understand the sentence, “Detached from monolayer culture eMSCs accumulate Na+ and have dissipated transmembrane ion gradients, while organized into compact spheroids, eMSCs restore the low Na+ content and the high K/Na ratio characteristic of functionally active cells”. Also, “sever” in the abstract should probably be “severe” and “constantly” in line 213 should also be something else.
Maybe shorten the Conclusions and move some of the text to Discussion
Reviewer 2 Report
1. MSCs characterization data must be presented.
2. Fig 1. There is no significant difference in ions at 2D and 3D conditions then why ion hemostasis is given so much importance that its mentioned in title of manuscript. Figure 1D and 1E need to be quantified.
3. Figure 2, Only cells without stain need to be included.
Why the cell population stain with DAPI is not shifted?
Quadrants are not uniform.
Graphical presentation for data must be included in this figure and should not be presented in a separate figure.
4. Figure 4D, standard deviation must be added to the bars.
5. Figure 5 should be combined with figure 1, or it must be figure 2.
6. Figure 6F, why control is significantly lower at 5 and 24h?
Round 2
Reviewer 2 Report
The corned raised in the first round of review are addressed in the revised version. The manuscript is accepted in its current form.